# Intrinsic anti-Stokes emission in living HeLa cells

**Laura Kacenauskaite**[ID]*, **Dovydas Gabrielaitis, Nicolai Bærentsen, Karen L. Martinez, Tom Vosch, Bo W. Laursen**[ID]

Nano-Science Center & Department of Chemistry, University of Copenhagen, Copenhagen, Denmark

* laura.kacenauskaite@chem.ku.dk

## Abstract

Intrinsic fluorescence of biological material, also called auto-fluorescence, is a well-known phenomenon and has in recent years been used for imaging, diagnostics and cell viability studies. Here we show that in addition to commonly observed auto-fluorescence, intrinsic anti-Stokes emission can also be observed under 560 nm or 633 nm excitation. The anti-Stokes emission is shown to be spatially located on/in the mitochondria. The findings presented here show that sensitive imaging experiments e.g. single molecule experiments or two-photon excitation imaging can be compromised if intracellular anti-Stokes emission is not accounted for. On the other hand, we suggest that this anti-Stokes emission could be exploited as an additional modality for mitochondria visualization and cell viability investigation even in systems that are already labeled with commonly used fluorophores that rely on normal Stokes-based detection.

## Introduction

Cell auto-fluorescence due to emission from intrinsic proteins, collagen, porphyrins, lipofuscin or adenine dinucleotides has been intensively investigated in the last decades and opened up possibilities for numerous applications like tissue, organelle or cell viability imaging [1–8]. The concentration distribution, pH, oxidation state of these molecules as well as temperature and polarity of the local environment were shown to influence fluorescence lifetime or/and quantum yield of naturally occurring fluorophores, making them useful intrinsic biomarkers in cells [4, 9–11]. Most commonly found in mitochondria, vesicles and extracellular matrix [4, 9, 12], these autofluorophores can help to shed a light on complex structures and functions of organelles without the need of introducing extrinsic fluorophores [13, 14].

In comparison to extrinsic fluorophores that are often used to specifically label various components of cells, the autofluorophores provide a direct relation between the emitter and its localization, while also ensure that neither the morphology nor the biological function are altered in the cells due to the introduction of external molecules. This advantage is utilized for cell viability studies, where changes in the mitochondria auto-fluorescence morphology are used to monitor cell stress and viability [15–17]. Additional applications are the label-free visualization of lysosomes [8, 18], chloroplasts [19] or tissue imaging [3, 7, 20]. Besides basic auto-

**Data Availability Statement:** All relevant data are within the manuscript and its Supporting Information files.

**Funding:** The author(s) received no specific funding for this work.

**Competing interests:** The authors have declared that no competing interests exist.

fluorescence intensity measurements, more advanced methods based on auto-fluorescence lifetime [20, 21], polarization [22] or transient state kinetics [23] are also being used in both imaging and diagnostics.

Despite the number of advantages outlined above, the intrinsic auto-fluorescence in biological samples also has some drawbacks. First, most of the auto-fluorescence imaging techniques require blue or near-UV light at relatively high intensities for exciting the auto-fluorescent molecules. A consequence of this is the loss of excitation selectivity as a myriad of molecules absorb in this wavelength region. Additional issues are photo-toxicity and photo-bleaching.

Here, we demonstrate that intrinsic emission of cells can also be detected on the anti-Stokes side, that is detected at wavelengths shorter than the excitation light. Conceptually, our imaging modality is similar to coherent two-photon absorption microscopy [5, 24] or upconversion microscopy using lanthanide based nanoparticles [25, 26]. Advantages of our method is that cell viability imaging can be done on a conventional, sufficiently sensitive fluorescence microscope (e.g. a confocal single molecule fluorescence microscope, equipped with an avalanche photodiode as detector); it does not require the introduction of external emitters and requires significantly lower excitation intensity in comparison to coherent two-photon absorption microscopy.

## Materials and methods

### Cell cultures

HeLa cells (Sigma-Aldrich, 93021013, obtained directly from supplier) were grown in culture medium in 37˚C, 5% $CO_2$ and >95% humidity. We used a culture medium composed of Dulbecco's Modified Eagle Medium Nutrient Mixture F-12 (DMEM/F-12) with GlutaMax$^{TM}$) (Gibco), supplemented with 10% (V:V) Fetal bovine serum (FBS–Gibco). In most of the cases, unless indicated otherwise, the culture medium containing 0.0159g/L phenol red, which is used as pH indicator.

During the last two days before imaging, the cells were cultured on glass coverslips in the same medium, at 37˚C, 5% $CO_2$. The surfaces were in all cases preliminary coated with Poly-L-Lysine (Sigma Aldrich) by incubating it in PBS (Sigma Aldrich) at 1:20 ratio (V:V) during 20 minutes.

To image stressed cells, we used transient short-cold stress conditions (incubation at 4˚C for 15 minutes), after which mammalian cells were shown to need several hours at physiological conditions to recover [27].

### Mitochondrial labeling

The labeling of mitochondria with MitoTracker™ Green FM (Thermofisher) was done by incubation of the cell sample in 100 nM MitoTracker™ Green FM solution for 20 min in 37 ˚C, 5% $CO_2$. After the incubation, the MitoTracker-Green FM solution was removed, and the cells were thoroughly washed (3 times with culture medium with phenol-red and 2 times with culture medium without phenol-red) before imaging the samples in culture medium without phenol-red.

### Cell imaging

Series of label-free and MitoTracker-Green stained cells were imaged using a setup previously described in ref. [28]. However, a few modifications were introduced: a 30/70 mirror was used instead of a dichroic mirror. FF02-485/20-25, LL01-514-25, LL02-561-25, and LL01-633-25 band-pass filters from Semrock were used to clean-up the excitation beam. An OlympusUPlanSApo

100x 1.4 NA oil immersion objective was used. In the detection path LP02-488RE-25, LP02-514RE-25, BLP01-561R-25, BLP01-633R-25, SP01-561RU-25 and BSP01-633R-25 filters from Semrock were used to block the corresponding excitation light. For the fluorescence images 400 nm, 485 nm, 510 nm and 560 nm excitation lines were selected from a pulsed continuum laser (77 MHz, 10 μW– 150 μW) SuperK Extreme EXB-6) with a SuperK SELECT wavelength selector from NKT Photonics. For the anti-Stokes imaging, a 561 nm Cobolt Jive from Cobolt (~1.3 mW before the objective) and 633 nm line of HRRR170-1 HeNe laser (~2.3 mW before the objective) from Thorlabs were used. Emission spectra were recorded by directing the emission light towards a spectrograph from Princeton Instruments (Princeton Instruments SPEC-10:100B/LN eXcelon CCD camera, SP 2356 spectrometer, 300 grooves/mm). Recorded anti-Stokes emission spectra contained significant fraction of scattering, which was eliminated by subtracting normalized reference spectra measured from the water drop on cover slip (see SI, S1 Fig). For anti-Stokes emission images BLP01-633R-25 and FF01-578/16-25 filters from Semrock were added in the detection path to suppress the scattering.

Cell images were recorded using an avalanche photodiode from PerkinElmer (CD3226) and sample scanning piezo stage (PI 517.3CL from Physik Instrumente) at controlled room temperature of 23˚C. The anti-Stokes emission of HeLa cells has been studied on ≈50 cells from 6 independently grown cell cultures throughout the course of half a year. The same signal localized in/on mitochondria has been detected in 100% of the healthy HeLa cells imaged and the anti-Stokes emission spectra recorded on 10% of them (randomly selected) were all peaked at around 590 nm.

All images were recorded as 90 x 90 μm images (256 x 256 pixels) with a scanning speed of 1ms/pixel and cropped for better visual representation after acquisition. Images were processed using Matlab and Fiji software.

## Results and discussion

Fig 1A to 1D shows auto-fluorescence images of label-free HeLa cells, cultivated in standard culture medium composed of DMEM, phenol-red and 10% FBS, at various excitation wavelengths. This figure demonstrates that normal auto-fluorescence (Stokes emission) from living HeLa cells can be detected over a large range of the visible spectrum, using a variety of excitation wavelengths. The spatial distribution of the auto-fluorescence signal indicates that the intrinsic fluorophores are not evenly distributed in the cytosol but predominately located in subcellular structures, most likely in/on mitochondria, as it was reported previously [1, 4]. Interestingly, when imaging the anti-Stokes emission, where we excite the cell sample at 560 nm or 633 nm and monitor emission at shorter wavelengths, images with similar spatial distributions are obtained (Fig 1E and 1F).

To confirm that both auto-fluorescence and anti-Stokes emission are originating from the mitochondria regions, HeLa cells were stained with MitoTracker-Green (excitation maximum ~490 nm, emission maximum ~510 nm), a well-established commercial fluorophore used for mitochondria identification in cells. Fig 2C shows that MitoTracker-Green fluorescence (Fig 2A) overlaps very well with the anti-Stokes emission (Fig 2B), confirming that the anti-Stokes emission originates from regions where the mitochondria are present. Differences in the composite images Fig 2C and Fig 2F correspond to movement of mitochondria in the living cells during the ~10 min time delay between the two imaging scans (auto-fluorescence and anti-Stokes, see also S2 Fig). Additionally, changing the excitation wavelength for the MitoTracker-Green stained HeLa cells to 560 nm (MitoTracker-Green does not absorb at this wavelength) yields again a good correlation between the auto-fluorescence and the anti-Stokes emission (see S3 Fig), proving that both signals originate from the same organelles.

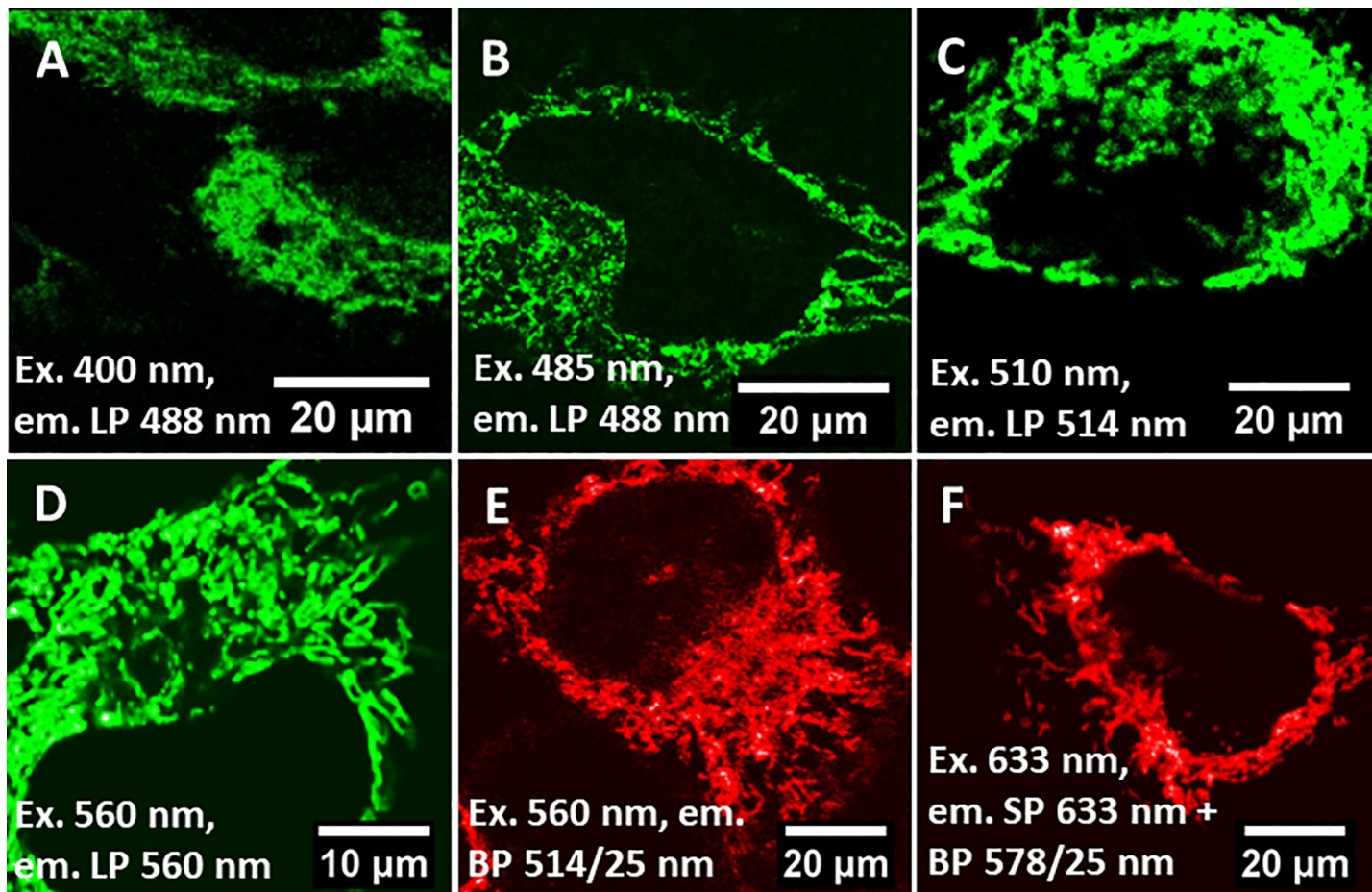

**Fig 1. Fluorescence images of label-free HeLa cells.** A to D) Auto-fluorescence upon excitation with 400 nm, 485 nm, 510 nm or 560 nm. E and F) Anti-Stokes emission upon excitation with 560 nm or 633 nm. The spectral detection ranges for both the auto-fluorescence and anti-Stokes emission are indicated in the images. Excitation wavelength (Ex.) long-pass filter (LP) band-pass filters (BP), short-pass filter (SP).

In order to start to understand the spectral properties of the anti-Stokes emission (Fig 2B and 2E) and to demonstrate the difference between auto-fluorescence (Fig 2D) and Mito-Tracker-Green emission (Fig 2A), emission spectra were recorded with a spectrometer coupled to the confocal microscope. Fig 3A shows that the fluorescence spectrum from the MitoTracker-Green stained cells was identical to the spectrum of pure MitoTracker-Green dissolved in water, with in both cases emission spectra peaking around 510 nm upon 485 nm excitation. Due to the high brightness of MitoTracker-Green, no significant contribution from the auto-fluorescence can be detected in the MitoTracker-Green stained HeLa cells. This can also be seen in Fig 3B, 3C and 3D where it is clear that auto-fluorescence in label-free HeLa cells only is observed at significantly higher excitation intensities, compared to the excitation intensities used to visualize the MitoTracker-Green. The auto-fluorescence spectrum is also clearly different from the MitoTracker-Green spectrum, with an emission maximum below 500 nm, as can be seen in Fig 3A. On the other hand, the anti-Stokes emission (ex. 633 nm) that is detected in the label-free HeLa cells has an emission maximum close to 590 nm. The two additional controls, exciting MitoTracker-Green stained cells and MitoTracker-Green alone in water, prove that the anti-Stokes emission does not originate from MitoTracker-

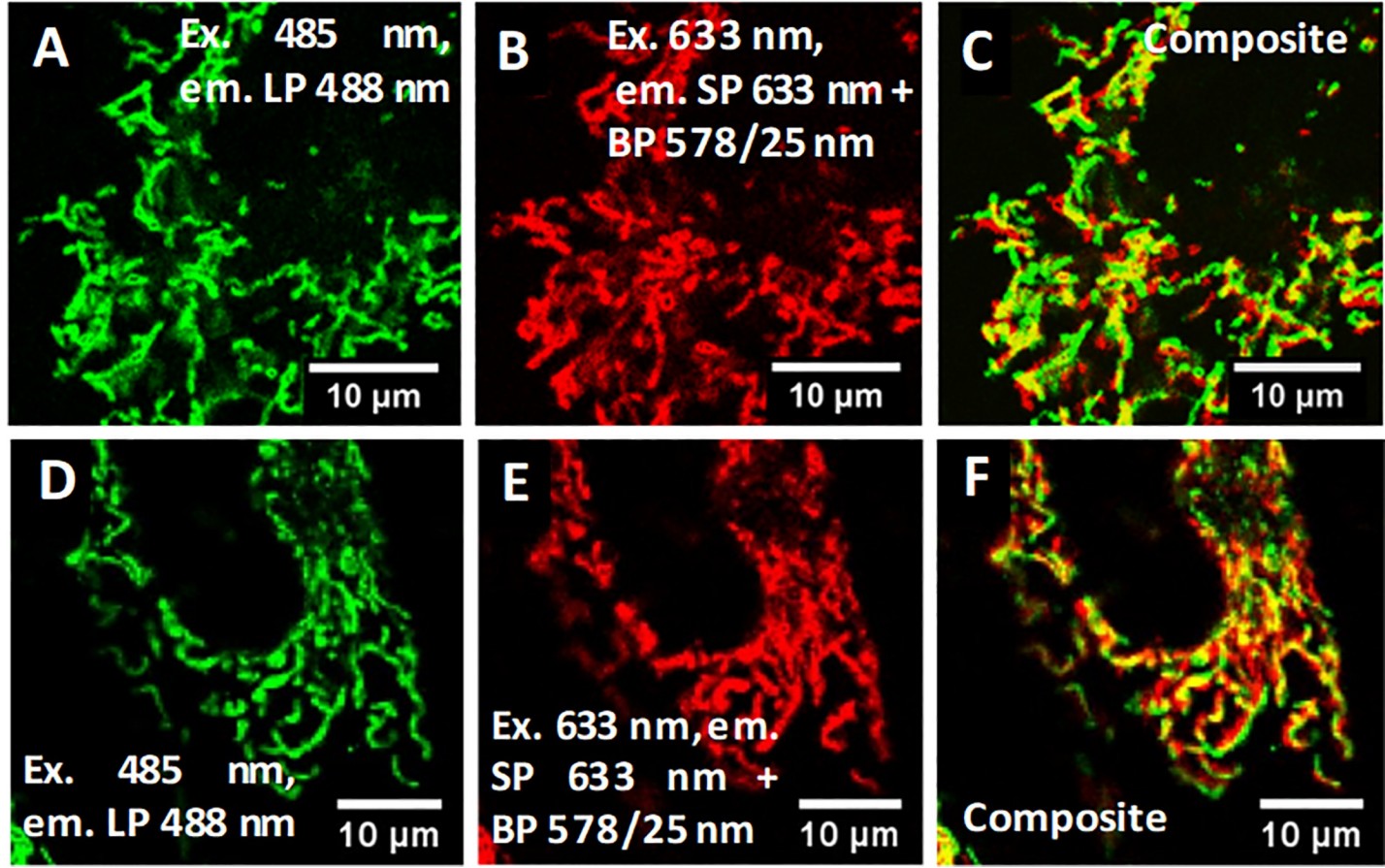

**Fig 2.** A) MitoTracker-Green fluorescence, (B) anti-Stokes emission and C) composite images of MitoTracker-Green stained HeLa cells. D) Auto-fluorescence, E) anti-Stokes emission and F) composite images of label-free HeLa cells.

Green and that presence of MitoTracker-Green in the HeLa cells has no significant impact on the spectral shape of the anti-Stokes emission.

Due to vast amount of different compounds present in living cells, it is not at all trivial to identify the compound(s) responsible for the observed anti-Stokes emission, let alone the mechanistic origin. Potential mechanisms that could explain the origin of the anti-Stokes emission are "hot band" excitation of thermally (Boltzmann) populated states [29], excited state annihilation (singlet or triplet) [30, 31], chemiluminescence [32] or consecutive photon absorption through a long lived intermediate state [33]. Coherent two-photon excitation can be ruled out as the intensity of the continuous wave laser used in our experiments was at least 1000 times lower than the peak intensity of pulsed Ti:Sapphire lasers often used for two-photon microscopy [34]. Additionally, with a potential two-photon excitation wavelength around 317 nm, one would expect a more blue shifted emission maximum, or at least some contribution from auto-fluorescence fluorophores in the blue part of the spectrum, which is not the case (Fig 3A). Due to the spatial location of the anti-Stokes emission, one could speculate that it is related to molecules localized in or on the mitochondria. However, the emission peak around 590 nm does not correspond to any of the commonly known endogenous auto-fluorescent compounds in the mitochondria [4, 9]. It would be beneficial to measure anti-Stokes emission excitation spectrum, lifetime and excitation power dependency to certainly rule out two photon absorption possibility, identify electronic transitions and narrow down on other

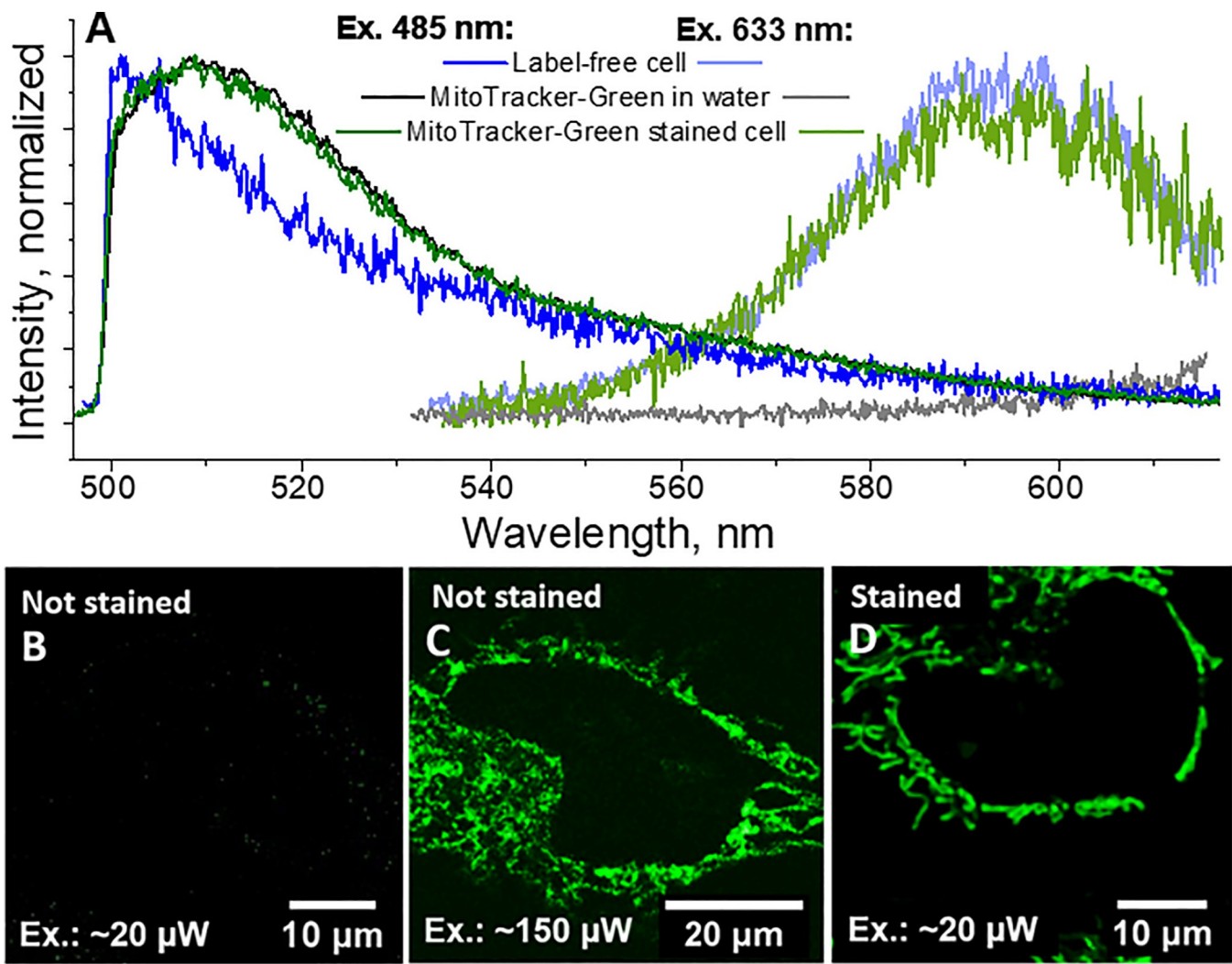

**Fig 3.** A) Normalized auto-fluorescence, MitoTracker-Green fluorescence and anti-Stokes emission spectra of MitoTracker-Green stained and label-free HeLa cells, (B to D) Differences in fluorescence intensity (ex. 485 nm, em. LP 488 nm) of MitoTracker-Green stained and label-free cells. The respective excitation powers are indicated in the figure. All figures have the same intensity scale range.

potential mechanisms. However, practically it is extremely challenging, as emission intensity is weak and photobleaching is too fast for these demanding experiments. Furthermore, identifying the exact molecule, let alone the mechanism, responsible for this anti-Stokes emission is also challenging not only because of technical limitations, but also due to a sheer number of potential autofluorophores and their complex photophysical behavior depending on configuration, oxidation state or response to specific microenvironment. Lack of possibilities to prepare and measure reliable negative controls in live cells then demand *ex vivo* 'deconstruction' of mitochondria molecule by molecule, which is tedious and at too time consuming at this point.

Here we speculate that the compounds, capable of emitting the anti-Stokes emission, associated with the mitochondria, are intrinsic to the cells. To verify this hypothesis, the only other potential source of the anti-Stokes emission, the cell growth medium, should also be investigated. Rich in proteins, sugars and other vital cell growth components, this medium also often contains additives that absorb light in the visible range, e.g. the pH indicator phenol red.

Despite its common use, little is known about interaction of phenol red with living cells. For a long time, it was assumed that this weakly fluorescent molecule does not interfere with any observable cell function, as its cellular uptake was considered minimal. However, modern quantification methods based on $^{125}$I labeled phenol red in cell culture medium showed that 0.31 picogram of phenol red can be accumulated per HeLa cell after just 2 hours of incubation in cell growth medium with ~10 µM phenol red [35]. Although S4 Fig shows that the phenol red containing medium by itself is able to give a weak emission with a slightly more red-shifted maximum around 610 nm, we do not believe that this is the origin of the anti-Stokes emission in the label-free HeLa cells. This is based on the fact that when HeLa cells were grown for 4 weeks (~30 division cycles) in phenol red-free media, similar anti-Stokes emission was observed as shown previously (see S4 Fig).

Here we propose to use this intrinsic anti-Stokes emission, from yet unknown compounds, to monitor cell stress in HeLa cells, even in fluorescently stained samples. As most of the auto-fluorophores are localized in mitochondria, autofluorescence is often used to evaluate cell stress/viability [36–39]. In case of extreme stress—cell death—membrane potential (including mitochondria membrane potential) is lost, membranes become permeable and previously localized molecules become free to diffuse throughout the cell volume.

This application is demonstrated in Fig 4, showing the well localized intrinsic anti-Stokes emission signal of healthy HeLa cells and its co-localization with the cell auto-fluorescence. Repetitive scans (Fig 4A–4F, approx. 6 min time between frames) of the same cell region showed minimal photobleaching and thereby confirm that the experimental conditions are not a relevant stress factor for the cells.

Upon stress, by incubation at 4˚C for 15 minutes, the well-localized anti-Stokes emission in the healthy HeLa cells (Fig 4A–4F) becomes completely delocalized over the whole cell volume (see Fig 4H). Similar as for the healthy cells, the auto-fluorescence and anti-Stokes emission remain co-localized (see Fig 4G–4I). This opens opportunities to study cell viability, stress and mitochondria shape using anti-Stokes emission.

The possibility to exploit anti-Stokes emission to obtain information about the stress level of cells has several advantages. First of all, the signal is intrinsic and does not require the addition of any fluorophores to the live cells. Its detection is furthermore done with distinct optical conditions (in comparison to regular cell dyes) which makes it compatible with many other fluorophores used in cells. It also enables the use of one more fluorophore/modality in multi-labeled samples and thereby provides more information about each cell composing the sample.

These features are very timely as anti-Stokes emission microscopy using upconverting fluorophores is becoming an important and highly studied modality in bioimaging that can be detected with any microscope setup using sensitive avalanche photodiodes.

## Conclusion

We have demonstrated that anti-Stokes emission from an intrinsic compound, associated with the mitochondria in HeLa cells, can be used as an additional channel for monitoring cell viability in both stained and unstained cells. Using red light excitation intensities, significantly lower than those used in coherent two-photon microscopy, yellow emission can be detected. The anti-Stokes emission signals overlap well with conventional auto-fluorescence from non-stained HeLa cells. Our results also demonstrate that attention must be paid to ensure that intrinsic anti-Stokes emission is accounted for, especially when using other anti-Stokes imaging modalities like upconversion emission from lanthanide based nanoparticles [40], optically-activated delayed fluorescence [33] or coherent two-photon absorption microscopy [24, 41].

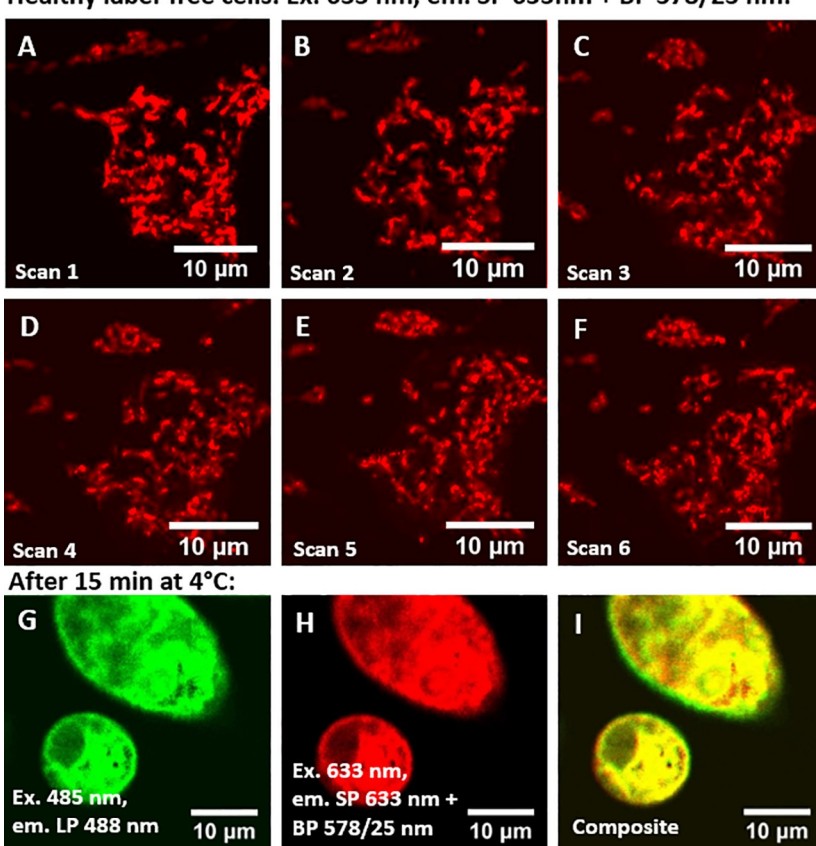

**Fig 4.** A-F) Multiple consecutive scans of healthy, label-free HeLa cells, G) auto-fluorescence, H) anti-Stokes emission and I) composite (images of stressed HeLa cells (15 minutes exposure to 4˚C).

## Supporting information

**S1 Fig. Example of high intensity scattering from cover slip (black line) being subtracted from anti-Stokes emission spectrum of unstained HeLa cells.**
(TIF)

**S2 Fig.** Two consecutive anti-Stokes emission scans (A and B) and their composite (C) (measured on non-stained HeLa cells, ~10 min delay between the scans), showing differences between the images arising due to cell movement.
(TIF)

**S3 Fig.** Auto-fluorescence (A), anti-Stokes emission (B) and composite (C) images of Mito-Tracker-Green stained HeLa cells. Note that at 560 nm MitoTracker-Green does not absorb and we are monitoring the intrinsic auto-fluorescence.
(TIF)

**S4 Fig.** Normalized anti-Stokes emission spectra (ex. 633 nm) of various samples compared to Stokes fluorescence (ex. 560 nm, orange spectrum) of phenol red (A). Cells here were grown in phenol red containing medium and exchanged to phenol red-free medium before imaging. Label-free HeLa cells, cultivated in phenol red-free medium for 4 weeks (B).
(TIF)

**S1 Datasets.**
(ZIP)

## Author Contributions

**Conceptualization:** Laura Kacenauskaite, Dovydas Gabrielaitis, Karen L. Martinez, Tom Vosch, Bo W. Laursen.

**Data curation:** Laura Kacenauskaite, Nicolai Bærentsen.

**Formal analysis:** Laura Kacenauskaite, Dovydas Gabrielaitis.

**Methodology:** Laura Kacenauskaite, Dovydas Gabrielaitis, Tom Vosch, Bo W. Laursen.

**Supervision:** Karen L. Martinez, Tom Vosch, Bo W. Laursen.

**Writing – original draft:** Laura Kacenauskaite, Tom Vosch.

**Writing – review & editing:** Dovydas Gabrielaitis, Nicolai Bærentsen, Karen L. Martinez, Tom Vosch, Bo W. Laursen.

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
