## [Decision Letter · Decision Letter 0]

21 Oct 2019

PONE-D-19-25352

Intrinsic anti-Stokes emission in living HeLa cells

PLOS ONE

Dear Ms. Kacenauskaite,

Thank you for submitting your manuscript to PLOS ONE. After careful consideration, we feel that it has merit but does not fully meet PLOS ONE’s publication criteria as it currently stands. Therefore, we invite you to submit a revised version of the manuscript that addresses the points raised during the review process.

We would appreciate receiving your revised manuscript by Dec 05 2019 11:59PM. To enhance the reproducibility of your results, we recommend that if applicable you deposit your laboratory protocols in protocols.io, where a protocol can be assigned its own identifier (DOI) such that it can be cited independently in the future. For instructions see: http://journals.plos.org/plosone/s/submission-guidelines#loc-laboratory-protocols

We look forward to receiving your revised manuscript.

Kind regards,

Debabrata Goswami

Academic Editor

PLOS ONE

Journal Requirements:

2. In your Methods section, please provide additional details regarding the cell lines used in your study and ensure you have described the source. For more information regarding PLOS' policy on materials sharing and reporting, see https://journals.plos.org/plosone/s/materials-and-software-sharing#loc-sharing-materials, and for more information on PLOS ONE's guidelines for research using cell lines, see https://journals.plos.org/plosone/s/submission-guidelines#loc-cell-lines.

Additional Editor Comments (if provided):

As noted by the reviewers, I do agree that the manuscript has merits to be further considered in PLoS One after their specific comments have been addressed. Overall, I feel the presentation and work in this paper is relevant and interesting. So, I hope the authors would agree to submit after addressing the major revisions for further consideration.

Reviewers' comments:

Reviewer's Responses to Questions

**Comments to the Author**

1. Is the manuscript technically sound, and do the data support the conclusions?

Reviewer #1: Yes

Reviewer #2: No

2. Has the statistical analysis been performed appropriately and rigorously? 

Reviewer #1: Yes

Reviewer #2: No

3. Have the authors made all data underlying the findings in their manuscript fully available?

Reviewer #1: Yes

Reviewer #2: Yes

4. Is the manuscript presented in an intelligible fashion and written in standard English?

Reviewer #1: Yes

Reviewer #2: Yes

5. Review Comments to the Author

Reviewer #1: The paper is coherent and describes an advancement in imaging techniques. The following minor grammar corrections are suggested:

- 31: "Concentration, distribution" -> "The concentration distribution"

- 38: "autofluorophores do not" -> "autofuorophores

Reviewer #2: The manuscript titled ‘Intrinsic anti-Stokes emission in living HeLa cells’ presents an observation of anti-stokes yellow emission from the mitochondria of HeLa cells. While this was an interesting observation to start with, unfortunately the authors overlooked existing literature and failed to verify the possible source and possible underlying mechanism of this emission. The manuscript is well written and data is presented concisely.

1) First of all, it should be mentioned that auto-fluorescence from mitochondria is well known and extensively studied. Mitochondria is the powerhouse of a cell and host an enormous amount of electron carriers. In literature, most of the auto-fluorescence has been assigned to electron carriers like NAD, NADP, FAD etc which shuttles between various emissive states depending on their oxidation status. As the author conclusively show that the emission originates in mitochondria, it’s advisable that the authors discuss their findings in this context throughout the paper.

2) Redox related anti-stokes emission from cellular auto-fluorescence has been reported by Melissa C. Skala and co-workers (PNAS 2007) and followed by many other groups. The 2-photon auto-fluorescence imaging is emerging as a powerful tool for optical redox imaging of cancer cells to predict malignancy. It should be noted that FAD emission occurs in the yellow region after exciting with a red laser. The fluorescence lifetime can be measured to differentiate between bound and un-bound states of these electron carriers. It’s recommended that the author to consider these literature for the best of their interest.

3) The authors argue that 590nm emission does not correspond to any know molecules but cite a very old reference (ref 9) which actually says cellular auto-fluorescence in the 500-600 nm spectral region is mostly associated with flavins. Although it’s agreeable that the exact maxima at 590nm is not well known and FAD emission is more blue-shifted from 590nm, but care must be taken to rule out flavins. Flavins are present in diverse form of molecules and oxidation states, especially in mitochondria and fluorescence is extremely sensitive to microenvironment. How many cells did the authors measure to confirm that 590nm emission is a ubiquitous in the mitochondria of HELA cell and there is no spectral shift from 590nm? Why did the authors choose 633nm? Where is the excitation maxima for that 590nm emission, is it 633nm? An excitation spectra would be valuable as it provides clue about the absorption and electronic states of the molecule of interest. Therefore it’s not convincing that authors rule out 2-photon excitation based on the their laser intensity.

4) The authors have randomly thrown in two possible mechanism of hot-band and dark states without any clear argument. Hot band emission should have an intense stokes component in red-region along with the anti-stokes yellow emission. A dark state emission is long-lived and should have a fluorescence lifetime an order higher in magnitude. Did the authors observe any of these?

5) The data presented in figure 4 is unclear and not explained. Are the scans shown in A-F is from the same ROI? If so, what was the time interval? Why the images looks significantly different from each other? Are the scan:A-D and scan:G-I from same ROI? Why the signal was delocalized, especially if it originates in mitochondria?

6) Note: ref 4 and ref 12 are same

In summary, the observation presented in the current manuscript is interesting. However the presented data is largely unexplained, looks preliminary and not fit for publication in its current state. Authors are encouraged to perform additional experiments to characterize the phenomena thoroughly, perform an extensive revision of their manuscript in the context of present literature, clearly present the research advancement and finally improve the scientific rigor of the data analysis and discussion for the interest of their future readers.

6. PLOS authors have the option to publish the peer review history of their article (what does this mean?). If published, this will include your full peer review and any attached files.

Reviewer #1: No

Reviewer #2: No

---

## [Author Response · Author response to Decision Letter 0]

12 Nov 2019

Reviewer 1: 

The paper is coherent and describes an advancement in imaging techniques. The following minor grammar corrections are suggested:

- 31: "Concentration, distribution" -> "The concentration distribution"

- 38: "autofluorophores do not" -> "autofuorophores

We are thankful for comments and positive feedback. We included these suggestions in a revised manuscript.

Reviewer 2: 

The manuscript titled ‘Intrinsic anti-Stokes emission in living HeLa cells’ presents an observation of anti-stokes yellow emission from the mitochondria of HeLa cells. While this was an interesting observation to start with, unfortunately the authors overlooked existing literature and failed to verify the possible source and possible underlying mechanism of this emission. The manuscript is well written and data is presented concisely.

1) First of all, it should be mentioned that auto-fluorescence from mitochondria is well known and extensively studied. Mitochondria is the powerhouse of a cell and host an enormous amount of electron carriers. In literature, most of the auto-fluorescence has been assigned to electron carriers like NAD, NADP, FAD etc which shuttles between various emissive states depending on their oxidation status. As the author conclusively show that the emission originates in mitochondria, it’s advisable that the authors discuss their findings in this context throughout the paper.

We are grateful for reviewer’s comments. Stokes-side autofluorescence in multiple cell organelles, including mitochondria, is indeed well-known and studied in detail. We briefly discussed most notable examples in the first two paragraphs of the manuscript. To stress the extensive research done specifically on electron carriers in mitochondria we added more references throughout the manuscript.

However, we also want to emphasize that with this paper we are presenting completely new way of mitochondria visualization using anti-Stokes autofluorophores. We strongly believe that this fluorescence is not due to two-photon absorption (see arguments in the following comment/answer), in contrary to majority of papers published on anti-Stokes autofluorescence to this day. Therefore, we believe that more extensive review of two-photon absorption based autofluorescence microscopy could be confusing and most importantly misleading to the readers.

2) Redox related anti-stokes emission from cellular auto-fluorescence has been reported by Melissa C. Skala and co-workers (PNAS 2007) and followed by many other groups. The 2-photon auto-fluorescence imaging is emerging as a powerful tool for optical redox imaging of cancer cells to predict malignancy. It should be noted that FAD emission occurs in the yellow region after exciting with a red laser. The fluorescence lifetime can be measured to differentiate between bound and un-bound states of these electron carriers. It’s recommended that the author to consider these literature for the best of their interest.

We acknowledged reviewer’s recommendation and added the reference to the manuscript.

While two-photon absorption induced emission of FAD was indeed well studied by multiple research groups (including paper by Melissa C. Skala) we believe and want to stress that anti-Stokes emission reported here is not due to coherent (simultaneous) two-photon absorption. 

First, as we argue in the manuscript (lines 172-177), the power of our continuous wave 633 nm HeNe laser line we are using for the excitation is at least three orders of magnitude lower than peak intensity of pulsed lasers commonly used in two photon absorption microscopy. Furthermore, with a potential two-photon excitation wavelength around 317 nm, one would expect a more blue shifted emission maximum, as 317 nm light is expected to efficiently excite well known and most abundant autofluorophores like NADH and FAD, with broad emission peaks around 460 and 540 nm, respectively. However, we did not observe any emission bellow 560 nm (Fig 3A in the manuscript), making two-photon absorption very unlikely hypothesis to explain the anti-Stokes emission we report.

3) The authors argue that 590nm emission does not correspond to any know molecules but cite a very old reference (ref 9) which actually says cellular auto-fluorescence in the 500-600 nm spectral region is mostly associated with flavins. Although it’s agreeable that the exact maxima at 590nm is not well known and FAD emission is more blue-shifted from 590nm, but care must be taken to rule out flavins. Flavins are present in diverse form of molecules and oxidation states, especially in mitochondria and fluorescence is extremely sensitive to microenvironment. How many cells did the authors measure to confirm that 590nm emission is a ubiquitous in the mitochondria of HELA cell and there is no spectral shift from 590nm? Why did the authors choose 633nm? Where is the excitation maxima for that 590nm emission, is it 633nm? An excitation spectra would be valuable as it provides clue about the absorption and electronic states of the molecule of interest. Therefore it’s not convincing that authors rule out 2-photon excitation based on the their laser intensity.

We performed more extensive literature review on emission spectra of various autofluorophores including more recent publications. However, to the best of our knowledge, no autofluorophores (including varying oxidation states) with the emission peak around 590 nm are reported so far. 

Even though flavins are most often associated with emission around 500-600 nm, number of other intrinsic and extrinsic molecules (for example cytochromes, lipopigments or phenol red) were reported to be emissive in 550-600 nm range. Furthermore, even NADH or metalloproteins cannot be safely ruled out, as their photophysical properties were reported to be highly environment dependent. In the manuscript we put special care to rule out extrinsic fluorophores like phenol red, as its contribution can be easily verified. However, finding the exact type of molecules responsible for anti-Stokes emission reported here becomes very difficult when it comes to intrinsic fluorophores. Difficulty comes not only from the sheer number of potential autofluorophores or lack of negative controls, but also complex photophysical behavior depending on configuration, oxidation state or response to specific microenvironment. Furthermore, we believe that more than one species could be responsible for observed anti-Stokes emission as anti-Stokes emission originating in mitochondria also can be observed using 560 nm excitation (Figure 1E), making identification of specific autofluorophores even more complicated.

Therefore, for now in this paper we chose to focus on applications and potential artefacts rather than identify exact type of molecule responsible for this emission.

We did not do an elaborate statistical analysis on spectral shape and peak position of observed emission as it wasn’t the scope of our work and the overall Anti-stokes emission is dim and subject to photobleaching, limiting the quality of the emission spectra and hence the analysis of emission maxima and other spectral properties. However, we measured multiple anti-Stokes emission spectra on live, non-stained HeLa cells, focusing on different cells and their mitochondria. We observed minimal spectral variations between different mitochondria and cells.

It would certainly be beneficial to measure anti-Stokes emission excitation spectrum, lifetime and excitation power dependency to certainly rule out two photon absorption possibility, but it is practically extremely challenging. Photobleaching is the biggest problem, limiting the possibility to measure excitation spectra, lifetimes and excitation power dependencies. We added these arguments to the manuscript.

4) The authors have randomly thrown in two possible mechanism of hot-band and dark states without any clear argument. Hot band emission should have an intense stokes component in red-region along with the anti-stokes yellow emission. A dark state emission is long-lived and should have a fluorescence lifetime an order higher in magnitude. Did the authors observe any of these?

When it comes to an origin of observed anti-Stokes emission, multiple photophysical pathways could be considered, including, but not limited to “hot band” excitation of thermally (Boltzmann) populated states, excited state annihilation (singlet or triplet) or consecutive photon absorption through a long lived intermediate state. We also added chemiluminescence (line 171 in the manuscript (1)) as another plausible route. 

So far we showed that it’s not likely to be due to two-photon absorption (see previous comments for the arguments). To address “hot band” absorption, we have measured emission on the Stokes side (as suggested by the reviewer) and observed weak, broad and featureless emission spectrum. Even though observed tail of emission spectrum could be a strong argument in case of a sample with single type of fluorophores, it is not conclusive in mitochondria, where multiple autofluorophores have been reported.

Other possible mechanisms could be investigated in more detail by analyzing Stokes and anti-Stokes emission lifetimes. However, as we pinpoint in the previous comment, it is not practically possible to record it due to extremely low fluorescence intensity and fast bleaching.

Since we are physically limited by the number of photons that can be recorded before bleaching, in this paper we only put an effort to show that mechanism responsible for this anti-Stokes emission is different than commonly reported 2-photon absorption. 

5) The data presented in figure 4 is unclear and not explained. Are the scans shown in A-F is from the same ROI? If so, what was the time interval? Why the images looks significantly different from each other? Are the scan:A-D and scan:G-I from same ROI? Why the signal was delocalized, especially if it originates in mitochondria?

Micrographs in Figure 4 A-F show consecutive scans of live, label-free HeLa cells and represent the same ROI. As we are using APD for detection instead of camera, it takes around 6 minutes to complete the scan. Meanwhile non-fixed, live cells and their organelles were observed to move (S2 Fig. in SI) significantly. We have now specified these parameters in the revised manuscript (highlighted additions at page 10). 

G-I scans correspond to different ROI than A-F, but rather show different distribution of anti-Stokes emitting molecules in the cell after cell stress was induced. Multiple reports have shown that cellular response to stress can be investigated following mitochondria dynamics – fission/fusion. As most of the autofluorophores are localized in mitochondria, autofluorescence are often used to evaluate cell stress/viability (2-5). In case of extreme stress - cell death - membrane potential (including mitochondria membrane potential) is lost, membranes become permeable and previously localized molecules become free to diffuse throughout the cell volume. To make these points clearer we have now added more discussion on cell viabilityon page 10 line 202-209.

Figure 4 demonstrates one of the main advantages of the method we are reporting here, showing that cell viability can be followed by looking at anti-Stokes emission. That not only allow investigate label-free cell viability without use of intense UV irradiation but also allows observation of cellular response to stress in stained samples. This wasn’t the case previously as Stokes autofluorescence is always buried under significantly more intense emission of external fluorophore in labeled cells.

6) Note: ref 4 and ref 12 are same

References have been corrected.

In summary, the observation presented in the current manuscript is interesting. However the presented data is largely unexplained, looks preliminary and not fit for publication in its current state. Authors are encouraged to perform additional experiments to characterize the phenomena thoroughly, perform an extensive revision of their manuscript in the context of present literature, clearly present the research advancement and finally improve the scientific rigor of the data analysis and discussion for the interest of their future readers.

We share the wish of the reviewer to fully understand the molecular and mechanistic origin of this new emission. However, we also strongly believe that this first observation of the phenomena and its potential applications merits publication, even though these academically interesting questions are not yet resolved. We hope that our reply and changes to the manuscript have satisfactory addressed the reviewers’ questions.

1. Ciscato LFML, Weiss D, Beckert R, Bastos EL, Bartoloni FH, Baader WJ. Chemiluminescence-based uphill energy conversion. New J Chem. 2011;35(4):773-775.

2. Westermann B. Bioenergetic role of mitochondrial fusion and fission. Biochimica et Biophysica Acta (BBA) - Bioenergetics. 2012;1817(10):1833-1838.

3. Youle RJ, van der Bliek AM. Mitochondrial fission, fusion, and stress. Science. 2012;337(6098):1062-1065.

4. Aubin JE. Autofluorescence of viable cultured mammalian cells. Journal of Histochemistry & Cytochemistry. 1979;27(1):36-43.

5. Dittmar R, Potier E, van Zandvoort M, Ito K. Assessment of cell viability in three-dimensional scaffolds using cellular auto-fluorescence. Tissue engineering Part C, Methods. 2012;18(3):198-204.

---

## [Decision Letter · Decision Letter 1]

5 Dec 2019

PONE-D-19-25352R1

Intrinsic anti-Stokes emission in living HeLa cells

PLOS ONE

Dear Ms. Kacenauskaite,

Thank you for submitting your manuscript to PLOS ONE. After careful consideration, we feel that it has merit but does not fully meet PLOS ONE’s publication criteria as it currently stands. Therefore, we invite you to submit a revised version of the manuscript that addresses the points raised during the review process.

The main point of concern of the paper that was raised by one of the referees is the important issue of the statistical significance of the new spectral features and that part is being brushed aside. 

We would appreciate receiving your revised manuscript by Jan 19 2020 11:59PM. To enhance the reproducibility of your results, we recommend that if applicable you deposit your laboratory protocols in protocols.io, where a protocol can be assigned its own identifier (DOI) such that it can be cited independently in the future. For instructions see: http://journals.plos.org/plosone/s/submission-guidelines#loc-laboratory-protocols

We look forward to receiving your revised manuscript.

Kind regards,

Debabrata Goswami

Academic Editor

PLOS ONE

Journal Requirements:

Additional Editor Comments

The authors seem to not interested to address the main concern of one of the reviewers on the lack of data and statistical significance of the work in their revised submission. Concern on the confidence level on new spectral signature need to be justified before the paper may be published.

Reviewers' comments:

Reviewer's Responses to Questions

**Comments to the Author**

1. If the authors have adequately addressed your comments raised in a previous round of review and you feel that this manuscript is now acceptable for publication, you may indicate that here to bypass the “Comments to the Author” section, enter your conflict of interest statement in the “Confidential to Editor” section, and submit your "Accept" recommendation.

Reviewer #1: All comments have been addressed

Reviewer #2: (No Response)

2. Is the manuscript technically sound, and do the data support the conclusions?

Reviewer #1: Yes

Reviewer #2: Partly

3. Has the statistical analysis been performed appropriately and rigorously? 

Reviewer #1: Yes

Reviewer #2: No

4. Have the authors made all data underlying the findings in their manuscript fully available?

Reviewer #1: Yes

Reviewer #2: Yes

5. Is the manuscript presented in an intelligible fashion and written in standard English?

Reviewer #1: Yes

Reviewer #2: Yes

6. Review Comments to the Author

Reviewer #1: (No Response)

Reviewer #2: The authors have partly addressed the comments previously made. Although the manuscript has been improved, the study appears to have limited impact in its current state. The authors argued that the signal is too weak and photo-bleaching too high to perform statistical analysis, lifetime measurement or even to identify the excitation maximum. This argument would raise a doubt about the significance or usefulness of this newly observed anti-stokes emission in this field, where other reported emissions are considerably strong. It would be advisable to take more time to perform additional experiments to characterize the new emission, thus, making the paper more valuable for the readers.

7. PLOS authors have the option to publish the peer review history of their article (what does this mean?). If published, this will include your full peer review and any attached files.

Reviewer #1: No

Reviewer #2: No

---

## [Author Response · Author response to Decision Letter 1]

8 Jan 2020

Reviewer #2: The authors have partly addressed the comments previously made. Although the manuscript has been improved, the study appears to have limited impact in its current state. The authors argued that the signal is too weak and photo-bleaching too high to perform statistical analysis, lifetime measurement or even to identify the excitation maximum. This argument would raise a doubt about the significance or usefulness of this newly observed anti-stokes emission in this field, where other reported emissions are considerably strong. It would be advisable to take more time to perform additional experiments to characterize the new emission, thus, making the paper more valuable for the readers.

The main concern of reviewer #2 in this second review is the question of impact or significance of our results based on the argument that if the signal is too weak to measure fluorescence lifetime or excitation spectrum, the signal might not be significant and thus relevant to be investigated. 

We disagree with this reasoning. As clearly shown in the data of the manuscript, the mitochondria signal resulting from this new anti-Stokes imaging is unambiguous, reproducible and can even be used to visualize cell stress. 

Anti-Stokes emission microscopy using upconverting fluorophores can be used today by anyone with a microscope setup equipped with sensitive avalanche photodiodes and is thus becoming an important and highly studied modality in bioimaging. We believe that it is important to report intrinsic anti-Stokes emission from the HeLa cells in the range that so far was always considered to be completely “background-free”, in particular when this signal is informative by indicating the stress level of the cell.

To emphasize the significance of reported anti-Stokes emission we added the following to the manuscript:

The possibility to exploit anti-Stokes emission to obtain information about the stress level of cells has several advantages. First of all, the signal is intrinsic and does not require the addition of any fluorophores to the live cells. Its detection is furthermore done with distinct optical conditions (in comparison to regular cell dyes) which makes it compatible with many other fluorophores used in cells. It enables the use of one more fluorophore/modality in multi-labeled samples and thereby provides more information about each cell composing the sample. 

These features are very timely as anti-Stokes microscopy using upconverting fluorophores is becoming an important and highly studied modality in bioimaging that can be detected with any microscope setup using sensitive avalanche photodiodes.

We share the wish of the reviewer to fully understand the molecular and mechanistic origin of this new emission and we agree that the lifetime and excitation spectra measurements, that reviewer #2 wish for, could be a good complement to our results from the fundamental point of view. However, getting these results requires very different instrumentation and emission intensities than what is required for imaging. We admit that we are not able to obtain these data as we depend on and are physically limited by the instrumentation that is available. 

Indeed, to record a reliable excitation spectrum of observed anti-Stokes emission, extremely powerful whitelight laser/lamp is necessary (output power of at least 10 mW per excitation width can be estimated based on our results and experience with 633 nm line of HeNe laser). We have previously attempted to record an excitation spectra using SuperK ExtremeEXB-6 with Super K SELECT wavelength selector from NKT Photonics, but no signal was observed due to too low excitation power (less than 1mW per excitation width). SuperK Extreme EXR-20 supercontinuum laser produced by the same company (which to our knowledge is the brightest commercially available whitelight laser on market to this day), also would not be sufficient as it “only” reaches ~3-7 mW/nm. That leaves us with only a few very specialized places in the world where such experiment could be carried out. Same technical limitations, together with fast bleaching, also apply to fluorescence lifetime measurements.

Furthermore, we also do not have hope that recorded excitation spectra could actually help to identify the molecule, let alone the mechanism, responsible for this anti-Stokes emission. As it was previously also noted by the reviewer, difficulty comes not only from the sheer number of potential autofluorophores but also complex photophysical behavior depending on configuration, oxidation state or response to specific microenvironment. Lack of possibilities to prepare and measure reliable negative controls in live cells then demand ex vivo ‘deconstruction’ of mitochondria molecule by molecule, which is practically impossible. Fluorescence lifetime measurements maybe could show if triplet state formation is involved in the process, but otherwise also in practice would not be helpful to identify anti-Stokes emitting species.

Editor: The main point of concern of the paper that was raised by one of the referees is the important issue of the statistical significance of the new spectral features and that part is being brushed aside. 

The authors seem to not interested to address the main concern of one of the reviewers on the lack of data and statistical significance of the work in their revised submission. Concern on the confidence level on new spectral signature need to be justified before the paper may be published.

The second point raised by both reviewer and the editor concerns statistical significance of our work. In the first review reviewer #2 specified this questions as: “How many cells did the authors measure to confirm that 590nm emission is a ubiquitous in the mitochondria of HELA cell and there is no spectral shift from 590nm”.

To ensure that observed intrinsic anti-Stokes emission is a general property of HeLa cells we imaged multiple cells from different batches under the same conditions. We have in this 2nd revision added further details on the number of cells and cell cultures investigated in the study. The following text has been added (and is highlighted in the submitted manuscript): 

The anti-Stokes emission of HeLa cells has been studied on �50 cells from 6 independently grown cell cultures throughout the course of half a year. The same signal localized in/on mitochondria has been detected in 100% of the cells imaged and the anti-Stokes emission spectra recorded on 10 % of them (randomly selected) were all peaked at around 590 nm. 

While this number of samples and spectra certainty is insufficient to derive any statistical significant quantitative statements e.g. of intensity of the emissions as function of cell conditions, we find it sufficient to support the qualitative scope of the paper: 1) reporting a new anti-Stokes imaging modality for mitochondria in HeLa cells and 2) drawing attention to this emission for other researchers applying imaging modalities for which this emission can interfere. 

We hope that our reply and changes to the manuscript have satisfactory addressed the reviewers’ and editors concerns.

---

## [Decision Letter · Decision Letter 2]

5 Feb 2020

PONE-D-19-25352R2

Intrinsic anti-Stokes emission in living HeLa cells

PLOS ONE

Dear Ms. Kacenauskaite,

Thank you for submitting your manuscript to PLOS ONE. After careful consideration, we feel that it has merit but does not fully meet PLOS ONE’s publication criteria as it currently stands. Therefore, we invite you to submit a revised version of the manuscript that addresses the points raised during the review process.

In particular, address the couple of more revision of the manuscript as per the comments of the Reviewer#2.  This would greatly enhance the article and we would be happy to consider the revised version favorably.

We would appreciate receiving your revised manuscript by Mar 21 2020 11:59PM. To enhance the reproducibility of your results, we recommend that if applicable you deposit your laboratory protocols in protocols.io, where a protocol can be assigned its own identifier (DOI) such that it can be cited independently in the future. For instructions see: http://journals.plos.org/plosone/s/submission-guidelines#loc-laboratory-protocols

We look forward to receiving your revised manuscript.

Kind regards,

Debabrata Goswami

Academic Editor

PLOS ONE

Reviewers' comments:

Reviewer's Responses to Questions

**Comments to the Author**

1. If the authors have adequately addressed your comments raised in a previous round of review and you feel that this manuscript is now acceptable for publication, you may indicate that here to bypass the “Comments to the Author” section, enter your conflict of interest statement in the “Confidential to Editor” section, and submit your "Accept" recommendation.

Reviewer #2: (No Response)

2. Is the manuscript technically sound, and do the data support the conclusions?

Reviewer #2: Yes

3. Has the statistical analysis been performed appropriately and rigorously? 

Reviewer #2: Yes

4. Have the authors made all data underlying the findings in their manuscript fully available?

Reviewer #2: Yes

5. Is the manuscript presented in an intelligible fashion and written in standard English?

Reviewer #2: Yes

6. Review Comments to the Author

Reviewer #2: The authors have improved the manuscript in this version. With the following minor revisions, it may be recommended for publication and further review may not be required.

1) It seems that the authors are physically limited by avilable facility, which often happens with scientific studies. However, if those inteded experiments can reveal imprtant clue about the emission, it needs to be discussed in the manuscript as a limitaion of the current study and possible future experiments for others in the field.

2) Why 4C, 15m was chosen for cell stress experiment may be briefly explained in the result/method section.

7. PLOS authors have the option to publish the peer review history of their article (what does this mean?). If published, this will include your full peer review and any attached files.

Reviewer #2: No

---

## [Author Response · Author response to Decision Letter 2]

20 Feb 2020

Reply to editor and reviewers:

Reviewer #2: The authors have improved the manuscript in this version. With the following minor revisions, it may be recommended for publication and further review may not be required.

We are grateful for Reviewer’s#2 thorough suggestions and comments towards inproved manuscript. Reviewer comments were considered in detail and addressed bellow:

1) It seems that the authors are physically limited by avilable facility, which often happens with scientific studies. However, if those inteded experiments can reveal imprtant clue about the emission, it needs to be discussed in the manuscript as a limitaion of the current study and possible future experiments for others in the field.

We have briefly mentioned the current limitations for further spectoscopic measurements in lines 190-191 and added more details in the revised manuscript.

2) Why 4C, 15m was chosen for cell stress experiment may be briefly explained in the result/method section.

We added a short explanation on cell stress in Materials and methods section.

We hope that our reply and changes have satisfactory addressed the reviewers’ comments and manuscript can now be accepted for publication.

---

## [Decision Letter · Decision Letter 3]

2 Mar 2020

Intrinsic anti-Stokes emission in living HeLa cells

PONE-D-19-25352R3

Dear Dr. Kacenauskaite,

We are pleased to inform you that your manuscript has been judged scientifically suitable for publication and will be formally accepted for publication once it complies with all outstanding technical requirements.

With kind regards,

Debabrata Goswami

Academic Editor

PLOS ONE

Additional Editor Comments (optional):

Reviewers' comments:

Reviewer's Responses to Questions

**Comments to the Author**

1. If the authors have adequately addressed your comments raised in a previous round of review and you feel that this manuscript is now acceptable for publication, you may indicate that here to bypass the “Comments to the Author” section, enter your conflict of interest statement in the “Confidential to Editor” section, and submit your "Accept" recommendation.

Reviewer #2: All comments have been addressed

2. Is the manuscript technically sound, and do the data support the conclusions?

Reviewer #2: (No Response)

3. Has the statistical analysis been performed appropriately and rigorously? 

Reviewer #2: (No Response)

4. Have the authors made all data underlying the findings in their manuscript fully available?

Reviewer #2: (No Response)

5. Is the manuscript presented in an intelligible fashion and written in standard English?

Reviewer #2: (No Response)

6. Review Comments to the Author

Reviewer #2: (No Response)

7. PLOS authors have the option to publish the peer review history of their article (what does this mean?). If published, this will include your full peer review and any attached files.

Reviewer #2: No

---

## [Editor Report · Acceptance letter]

4 Mar 2020

PONE-D-19-25352R3 

Intrinsic anti-Stokes emission in living HeLa cells 

Dear Dr. Kacenauskaite:

I am pleased to inform you that your manuscript has been deemed suitable for publication in PLOS ONE. Congratulations! Your manuscript is now with our production department. 

With kind regards,

on behalf of

Dr. Debabrata Goswami 

Academic Editor

PLOS ONE